# Gestational Diabetes—Placental Expression of Human Equilibrative Nucleoside Transporter 1 (hENT1): Is Delayed Villous Maturation an Adaptive Pattern?

**DOI:** 10.3390/diagnostics13122034

**Published:** 2023-06-12

**Authors:** Cinzia Giacometti, Kathrin Ludwig, Monica Guidi, Elvira Colantuono, Anna Coracina, Marcello Rigano, Mauro Cassaro, Alessandro Ambrosi

**Affiliations:** 1Pathology Unit, Department of Diagnostic Services, ULSS 6 “Euganea”, 35131 Padova, Italy; mauro.cassaro@aulss6.veneto.it; 2Pathology Unit, Department of Medicine, University of Padova, 35128 Padova, Italy; kathrin.ludwig@aopd.veneto.it; 3Gynecology & Obstretics Unit, Department of Women’s Health, Cittadella Hospital, ULSS 6 “Euganea”, 35013 Padova, Italy; monica.guidi@aulss6.veneto.it; 4Gynecology & Obstretics Unit, Department of Women’s Health, Camposampiero Hospital, ULSS 6 “Euganea”, 35012 Padova, Italy; elvira.colantuono@aulss6.veneto.it (E.C.); marcello.rigano@aulss6.veneto.it (M.R.); 5Diabetology Unit, Department of Medicine, Camposampiero Hospital, ULSS 6 “Euganea”, 35012 Padova, Italy; anna.coracina@aulss6.veneto.it; 6School of Medicine, Vita-Salute San Raffaele University, 20132 Milano, Italy; ambrosi.alessandro@hsr.it

**Keywords:** gestational diabetes, placenta, hENT1, vasculo-syncytial membrane, Ki-67, p57

## Abstract

Gestational diabetes mellitus (GDM) is a metabolic disease that can affect placental villous maturation and villous vascularity. The main effects of GDM on placental growth are a delay of villous maturation (DVM) and decreased formation of vasculo-syncytial membranes (VSM). Human equilibrative nucleoside transporter-1 (hENT1) is an adenosine transporter expressed in the human umbilical vein endothelial cells (HUVEC) and human placental microvascular endothelium cells (hPMEC). Its role is crucial in maintaining physiological fetal adenosine levels during pregnancy, and its reduction has been described in GDM. Twenty-four placentas from pregnancies with a confirmed diagnosis of GDMd and twenty-four matched non-GDM placentas (controls) were retrospectively analyzed to investigate the immunohistochemical expression of hENT1 in HUVEC and hPMEC. The study included the quantitative evaluation of VSM/mm^2^ in placental tissue and the immunohistochemical quantitative evaluation of Ki-67, PHH3, and p57 in villous trophoblast. hENT1 expression was higher in all the vascular districts of the control cases compared to the GDMd placentas (*p* < 0.0001). The VSM/mm^2^ were lower in the GDMd cases, while the Ki-67, PHH3, and p57 were higher when compared to the control cases. To our knowledge, this is the first report of hENT1 expression in the human placentas of GDM patients. The absence/low expression of hENT1 in all the GDMd patients may indicate a potential role in microvascular adaptative mechanisms. The trophoblasts’ proliferative/antiapoptotic pattern (high Ki-67, high PHH3, and high p57 count) may explain the statistically significant lower number of VSM/mm^2^ found in the GDMd cases.

## 1. Introduction

Gestational diabetes mellitus (GDM) is a disease defined by its onset or first recognition during pregnancy and is characterized by glucose intolerance, leading to maternal hyperglycemia. Its incidence accounts for 15% of pregnancies in developed and developing countries [1]. Although GDM resolves after birth, it is associated with changes during prenatal life, perinatal alterations (e.g., macrosomia, insulin resistance, and higher systolic blood pressure), and diseases in adulthood (e.g., diabetes, obesity, dyslipidemia, hypertension, and metabolic syndrome) [2,3,4,5,6]. 

One pathophysiological consequence of GMD is altered vascular function, defined as the altered capacity of the endothelium to take up and metabolize the cationic amino acid L-arginine, the substrate for NO synthesis via NO synthases (NOS) [7,8]. Since the placenta lacks innervation, the physiological vascular placental function is maintained by locally released vasoactive molecules from the endothelium, such as the gas nitric oxide (NO) or the endogenous nucleoside adenosine [9]. Adenosine is a vasodilator in most vascular beds, including the human placenta. The main biological effects of adenosine are to maintain the homeostatic equilibrium and act as a key stimulator of angiogenesis. The latter results in increased L-arginine transport-dependent NO synthesis via the endothelial NO synthase. Thus, a functional link between adenosine and the endothelial L-arginine/NO pathway (ALANO pathway) has been proposed [9]. Human equilibrative nucleoside transporter-1 (hENT1) is an adenosine transporter expressed in human umbilical vein endothelial cells (HUVEC) and human placenta microvascular endothelium cells (hPMEC) [10,11]. In HUVEC, adenosine transport is mainly mediated by hENT1. HUVEC and hPMEC are known to be metabolically crucial in maintaining normal adenosine extracellular levels by efficient uptake of this nucleoside [12], thus modulating its broad biological effects. Differential expression of adenosine receptor subtypes is a factor known to contribute to the functional heterogeneity of human placental macro- and microvascular endothelium. Its role is crucial in maintaining physiological fetal adenosine levels in utero, and its reduction has been described in GDM pregnancies in cultured cells derived from GDM placentas and umbilical cords. This phenomenon is associated with a lower capacity of adenosine transport via human equilibrative nucleoside transporters (hENTs) by HUVEC and hPMEC in GDM [12]. The histological hallmark of the effects of GDM on placental maturation is the so-called delay of villous maturation (DVM), which also encompasses the decreased formation of vasculo-syncytial membranes, the presence of multiple centrally located capillaries and a variable extent of chorangiosis [13,14]. Placental villous maturation reaches the highest point in the 3rd trimester, with an abundance of terminal villi, defined by small-caliber vessels (40–100 µm), minimal stroma, and abundant vasculo-syncytial membrane (VSM) formation [15]. The VSM is the structure that must allow and facilitate the optimal gas exchange between the maternal blood lakes and the fetal bloodstream through the placental villi. The VSM derives from the “fusion” of fetal capillary walls, which are peripherally located, in strict proximity to the trophoblast basement membrane and an ultra-thin layer of trophoblast cytoplasm. At term, terminal villi account for 40–50% of the placental volume and 60% of the cross-sectional area [16]. Delayed villous maturation (DVM) is an entity by which the maturation of the terminal placental villi is abnormal, and it takes place to a lesser extent for gestational age [17]. The villi are usually large and enlarged, with a higher number of stromal cells and edematous stroma. Many capillaries are not peripherally located on higher-power magnification, resulting in a decrease in VSM formation. The trophoblast surrounding the villi appears thickened and hypercellular [18]. The placentas of GDM women are usually larger, thicker, and heavier compared to those of women with normal pregnancies. Even if the exact mechanisms accounting for the increased placental mass remain mainly unclear, some suggestions of alterations in trophoblast cells’ proliferation, differentiation, and cell death have been proposed. Some authors described an increase in cellular proliferation markers, detected by proliferative cell nuclear antigen (PCNA) and Ki-67, in the various villous cell types: cytotrophoblasts, syncytiotrophoblasts, stromal cells, and endothelial cells [19]. Although these changes may contribute to the well-known increased placental size, an alternative hypothesis suggests a possible cause in the dysregulation of trophoblast cell death [20]. Together with Ki-67, another marker of cell proliferation is the phosphorylation of histone H3 at serine10 (H3S10P). This is an important event in the cell cycle progression, starting in the pericentromeric chromatin in the late G2 phase. The process then spreads non-randomly throughout the condensing chromatin during the prophase, persisting throughout the anaphase of the cell cycle. As phosphohistone H3 (PHH3) phosphorylation is typically no longer detectable when mitosis is completed and is not expressed in apoptotic bodies, it is a specific marker of mitotic figures [21]. p57 is a protein of the CIP/KIP family of cyclin-dependent kinase (CDK) inhibitors (CKIs). In human placentas, it is present in the villous cytotrophoblasts, villous stromal cells, amniotic epithelium, invasive cytotrophoblasts, and decidual cells. Due to its role in cell cycle control, p57 is involved in regulating a variety of cellular processes, such as embryogenesis and tissue differentiation, and its regulation is highly complex. p57 has a highly specific profile expression, both spatially and temporally: its peak and widespread distribution are at a maximum during embryogenesis and development. At the same time, it remains restricted to only a few tissues in adult life [22]. 

This study aimed to investigate the immunohistochemical expression of hENT1 in HUVEC and hPMEC in delivered placentas of GDM patients in dietary treatment (GDMd). The study included the quantitative evaluation of the VSM, Ki-67, PHH3, and p57 and their correlation with hENT1 and fetal, placental, and maternal characteristics.

## 2. Materials and Methods

This retrospective observational study did not imply any change in therapeutic or diagnostic procedures. Only intact placentas and patients with available clinical data and analysis were considered in the study. Placentas from patients with a GDMd diagnosis and no other co-morbidities were consecutively collected in the ULSS 6 Community Hospitals of Camposampiero and Cittadella (Padua, Italy) from April to July 2018. A diagnosis of GDM was achieved in the presence of at least one glycemic level above the normal in the two-hour test with 75 g syrup glucose solution: equal to or higher than 92 mg/dL (5.11 mmol/L) immediately after (time 0) and/or equal to or higher than 180 mg/dL (10 mmol/L) after 60 min and/or equal to or higher than 153 mg/dL (8.5 mmol/L) after 120 min [1]. Control cases, with normal glucose tolerance tests during pregnancy, were collected from consecutive deliveries in the same period. All the delivered placentas were formalin-fixed and paraffine-embedded (FFPE) after 2–4 days of 4% buffered formalin fixation. Placental sampling was conducted according to the Amsterdam protocol [17], modified as follows: A total of six samples were collected from every placenta: one sample of membranes (membrane roll) and umbilical cord (proximal, intermediate, and distal, near to cord insertion), one sample including cord insertion, three samples of placental parenchyma (two center-parenchymal, one of the most and one of the least normal area, and one para-central sample. All the diagnoses were rendered according to the Amsterdam protocol, as follows: delayed villous maturation (DVM), maternal vascular malperfusion (MVM), chorioamnionitis, villitis of unknown etiology (VUE), or a combination of two (or more) diagnoses (MVM + DVM, MVM + chorioamnionitis, etc.). The mid-portion of the placental parenchyma was assessed in 1 section from each placenta. Four consecutive high-power fields (about 1 mm^2^; field diameter 0.62 mm; field area 0.302 mm^2^) were evaluated, and the VSM were counted in all the terminal villi present in each field [16]. The same slide used for the VSM evaluation was incubated with p57 (mouse monoclonal antibody, 5 mL dispenser, pre-diluted, ~1.3 µg/mL, Roche Diagnostics), Ki-67 (clone 30-9, rabbit monoclonal antibody, 5 mL dispenser, pre-diluted, ~2 µg/mL, Roche Diagnostics), and PHH3 (rabbit polyclonal antibody, 5 mL dispenser, pre-diluted, ~1.3 µg/mL, Cell Marque, Sigma Aldrich). Ki-67 and p57 were scored by counting the percentage of positive nuclei on 200 villous syncytiotrophoblast cells; PHH3 immunohistochemistry was scored by counting the positive nuclei/mm^2^ (as reported above) in villous syncytiotrophoblast cells. Umbilical cord and placental parenchyma samples from the GDMd cases and non-GDM controls were incubated with hENT-1 antibody (clone sp120, rabbit monoclonal antibody, pre-diluted, ~0.44 µg/mL, Roche Diagnostics); incubation without the primary antibody served as a negative control. According to the manufacturer’s instructions, the normal pancreas served as a positive control. As hENT1 immunohistochemical expression is not reported in the literature applied to the human placenta, its expression was quantitatively evaluated in the HUVEC and hPMEC using the proportion of positive cells and multiplying the percentage of cells demonstrating each intensity and adding the results, as for the $H-score=\overset{3}{\underset{i=0}{\sum }}i\cdot p_{i}$ described for the quantification of estrogen and progesterone receptor expression in the breast, where *i* is the intensity score (scored from 0, absent, to 3, strong, intense reaction) and $p_{i}$ is the corresponding percentage of the cell [23]. A score between 0 and 300 was achieved for each case and in each vascular district (HUVEC and hPMEC). The syncytiotrophoblast cells’ basement membranes served as the positive internal control.

### 2.1. Statistical Methods

#### 2.1.1. Pre-Processing

After a preliminary check, we used the propensity score to select samples of matching subjects to reduce the possible risk of bias due to confounding variables. We used logistic regression to estimate the propensity scores considering the following covariates: BMI and age at delivery, fetal/placental weight ratio (F/P), and placental/fetal weight ratio (P/F). We matched (1:1) the cases and controls using the optimal matching approach, which keeps the sum of the absolute pairwise distances in the matched sample as small as possible, without replacement. We assessed the balance in the distribution of the covariates before and after matching using the standardized mean difference and variance ratio (an absolute standardized mean difference of <0.2 after matching was considered to indicate a good balance) and checked the test power. The following analyses were applied to the matched samples.

#### 2.1.2. Analysis

The observed numeric values were summarized by the mean and standard deviation (SD), and the categorical values were summarized by frequencies and percentages. To allow an easier comparison with the literature, the complete descriptive statistics are reported in the Appendix A. The correlation between the numeric variables was computed by means of Spearman’s correlation coefficient, ρ. The difference in the means between the cases and controls was assessed by means of Welch’s t-type test. The dependencies between the categorical variables were assessed with Fisher’s test or the Chi-squared test.

To find subgroups of patients, we performed an unsupervised hierarchical cluster analysis based on the following: maternal variables (diastolic arterial pressure, systolic arterial pressure, BMI at delivery), fetal variables (APGAR score al 5′, fetal weight, gestational weeks), placental variables (umbilical cord index—UCI—defined as the number of umbilical vein coils/cm, placental diameters—maximum and minimum, placental weight, umbilical cord diameters—maximum and minimum, placental thickness—maximum and minimum), immunohistochemical variables (expression of p57, proliferation index scored by Ki-67, expression of hENT1 in PMEC, expression of hENT1 in HUVEC), and morphological variables (number of vasculo-syncytial membranes/mm^2^, VSM/mm^2^). The hierarchical clustering was based on Ward’s criterion with the Euclidean distance for the patients and the pairwise correlational distance ($1-\left|\rho \right|$) for the variables. We graphically represented the results as heatmaps with dendrograms obtained from both complete hierarchical clustering results. We applied a least absolute shrinkage and selection operator (LASSO) to a logistic regression model considering all the variables included in the cluster analyses to identify a minimum set of the most informative variables. The variables with a nonzero coefficient in the LASSO logistic model were selected as a set of potential predictors. Finally, we checked the capability to classify the GDMd versus non-GDM cases of the selected variables by an optimism-adjusted Sommer’s D index based on bootstrap (B = 20,000) and the associated classification accuracy. To further graphically inspect the capability of the selected variables to identify the two groups of patients, we plotted the first two dimensions of the Principal Component Analysis (PCA), along with the elliptic convex hull of the cases and controls. Exact *p*-values were computed by means of permutation methods to avoid any distributional approximation, and the significance level was set at α= 0.05. All the statistical analyses were performed with R (version 4.2.1).

## 3. Results 

A total number of 81 consecutive placentas were submitted to the Pathology Laboratory between April and July 2018, according to the guidelines [24]. Of these, two were excluded because of incomplete clinical or laboratory data and three were excluded due to fragmentation during delivery (manual removal of the afterbirth). All the remaining placentas (*n* = 76) fulfilled the inclusion/exclusion criteria: 28 placentas from diagnosed GDMd mothers (cases) and 48 placentas associated with non-GDM diagnoses (controls). After 1:1 propensity score matching, 24 GDM cases were matched with 24 non-GDM controls (6 intra-uterine growth restrictions, 6 cases of fetal distress during labor, 3 cases suspicious for placenta accreta, 2 placentas with small for gestational age fetuses, 2 cases of premature rupture of membranes, 1 pre-term delivery, 1 maternal fever during labor, 1 oligohydramnios, 1 case of hepatogestosis, 1 placenta previa). The resulting sample size allowed a test power of 1−β= 0.95 with respect to Cohen’s effect size d= 0.8 and a significance level α= 0.05.

The patient, newborn, and delivery characteristics are summarized in Table 1. 

According to the Amsterdam protocol, the histological diagnoses of GDMd cases consisted of DVM in 20 cases (83.3%) and DVM with focal features of MVM in 4 cases (16.7%). Among the control cases, 12 cases (50%) were diagnosed as MVM with focal features of DVM, 8 cases (33.3%) as MVM, 2 cases (8.3%) as normal placenta with infections (chorioamnionitis), 1 case (4.2%) as VUE, and 1 case (4.2%) as normal. As expected, the GDMd placentas were heavier (477.54 ± 75.96 g versus 394.91 ± 120.96 g, *p* = 0.0007) and the GDMd fetal weight was higher (3248.54 ± 501.07 versus 2626.12 ± 568.66 g, *p* < 0.0001) than in the non-GDM cases. There was no statistical difference between the two groups with regard to the BMI at term and ΔBMI. The mean expression of hENT1 in the control cases versus the GDMd placentas was 159.17 ± 54.76 versus 10 ± 23.03 in the hPMEC (*p* < 0.0001) and 194.17 ± 66.52 versus 11.25 ± 21.93 in the HUVEC (*p* < 0.0001) (Figure 1). 

The Ki-67, PHH3, and p57 counts were significantly higher in the GDMd group when compared to the controls (Ki-67 count 19.95 ± 2.83 versus 7.5 ± 2.37, *p* < 0.0001; PHH3 count 3.79 ± 2.26 versus 0.96 ± 0.80, *p* < 0.0001; p57 count 20.87 ± 4.72 vs 8.45 ± 2.84, *p* < 0.0001). This “proliferative”/antiapoptotic pattern (high Ki-67, PHH3, and p57 counts) seems to sustain the statistically significantly lower number of VSM/mm^2^ found in the GDMd cases compared to the controls (3.67± 2.96 versus 20.04 ± 8.77; *p* < 0.0001) (Figure 2). 

Bootstrap-based resampling gave an optimism-corrected Sommer’s D index of 1 and a mean accuracy equal to 100%. However, we underline that these results should be taken carefully, given the final sample size. The main results are summarized in Table 2 and Figure 3.

## 4. Discussion

The screening strategies for GDM used across Europe still bear many differences between countries. By applying selective screening according to European guidelines, approximately 50% of pregnant women would need to be subjected to a glucose tolerance test. The application of Dutch guidelines permits the reduction of the percentage of undiagnosed cases (“only” 33%) [25]. Insulin therapy could be clinically considered useful to restore normal maternal and fetal glycemia when dietary therapy is insufficient; however, its usefulness in avoiding endothelial dysfunction is still unclear [26]. This could be due, as previously demonstrated in in vitro models, to the incapacity of insulin therapy per se to restore fully normal vascular functionality in GDM placenta micro-macrovasculature [27]. Insulin stimulates fetal aerobic glucose metabolism and will increase the fetus’s oxygen demand. If adequate supply is not available due to reduced oxygen delivery (to the intervillous space because of the higher oxygen affinity of glycated hemoglobin, thickening of the placental basement membrane, and reduced uteroplacental or fetoplacental blood flow), fetal hypoxemia will ensue, despite therapy. Maternal diabetes has several effects on the human placenta. Characteristically, the placenta in GDM, as we also demonstrated in the present study, is heavier and thicker, with an enlarged surface area of exchange both on the maternal (syncytiotrophoblast) and fetal (endothelium) side. It may appear paradoxical that in a situation of maternal nutritional oversupply, the placenta increases its weight, thickness, and surface, thus potentially contributing to enhanced maternal-fetal transport. This kind of adaptation reflects both the crucial importance of guaranteeing an adequate oxygen supply to the fetus and the effect of excess growth factors (such as insulin), which drives some of the placental changes, even if they result in adverse side effects. If morphology may give a clue about the pathogenesis of structural placenta abnormalities, the molecular substrate may provide a possible answer, as clinical manifestations of GDM seem to rely on fetoplacental endothelial dysfunction [7]. Different mechanisms leading to vascular alterations present in GDM have been investigated. Among these, the reduced adenosine uptake due to reduced hENT1 transport capacity has been widely investigated [28,29]. Adenosine is an endogenous purine nucleoside formed in both intra- and extracellular spaces. The production of extracellular adenosine originates from the dephosphorylation of extracellular adenosine monophosphate via the nucleotidases displayed in the plasma membrane of endothelial cells [30], including HUVEC [31]. Adenosine plays many different roles in vascular tissues, such as the regulation of vascular tone and blood flow [10]. The abnormally elevated extracellular adenosine concentration described in the culture medium of HUVEC may lead to the activation of adenosine receptors, which could repress adenosine transport via hENT1 in hPMEC [27]. These could be crucial mechanisms in the maintenance of physiological extracellular adenosine levels both in the micro-and macro-circulation of the human placenta in GDM [32]. The efficiency in adenosine uptake has been reported to be altered in isolated HUVEC from GDMd patients, so it was postulated that a reduction in hENT1 likely occurred in the endothelial cells [9]. Endothelial dysfunction is a crucial aspect of GDM. In 2013, Pardo et al. [33] reviewed the role of adenosine and its receptors in GDM. In their exhaustive review, they highlighted the existing link between nitric oxide (NO) and the adenosine pathway in the so-called ALANO pathway [7]. According to this hypothesis, HUVEC and hPMEC from GDM patients result in higher levels of NO, a functional severance between NO synthase and L-arginine uptake, increased levels of adenosine, and reduced expression of hENT1 [33,34,35]. In our study, we investigated the presence of the immunohistochemical expression of hENT1 in different vascular districts in GDMd placentas and non-GDM placentas. We demonstrated that hENT1 membrane expression is largely diminished or absent in GDMd placentas, while it remains largely unaltered in all other cases. This phenomenon has been observed both in HUVEC and hPMEC, compared to control cases with normal basal glycemia. In non-GDMd cases, hENT1 expression was moderate to strong and diffusely present in all the vascular districts investigated; for this reason, the absence/reduction of hENT1 expression is likely to identify potential GDMd placentas, even in the absence of clinically/serological proven GDM. One of the theses postulated in this paper is that the peculiar morphological aspect of placental tissue (larger, “edematous”, hypervascularized, and immature villi) in GDM patients (the so-called DVM) [17] is the expression of an adaptative mechanism apt to reduce fetal exposure to the highly oxidative environment caused by hyperglycemia and hyperinsulinism rather than a true immatureness of the villous population. Hyperglycemia and hyperinsulinemia of diabetic pregnancies may be one of several mechanisms by which DVM occurs, but in a previous study, no differences between glycosylated hemoglobin or fructosamine values in pregestational diabetic patients with or without DVM were identified [36]. Various etiologies for the pathogenesis of DVM have been suggested, including increased levels of placental growth factors, as seen in diabetes and maternal obesity [37]. At the microscopic level, the specific histological feature of GDM is DVM, with rates in diabetic placentas ranging from 81% [38] to 16.6% [18]. We assume that DVM could be a misnomer, as the villi in GDM patients are not truly immature, even if they display a morphology similar to the villi from the 33rd to 34th weeks of gestation. In our opinion, the lack of formation of the VSM might be a protective mechanism in GDM placentas rather than an unwanted side effect of hyperglycemia and hyperinsulinemia. The inadequate number of VSM in the terminal villi of GDM placentas, probably due to the higher proliferation of villous syncytiotrophoblasts (higher Ki-67 value and mitosis number), creates a physical barrier between the intervillous maternal blood and fetal villous capillaries, thickening the virtual space of the VSM by the proliferation of trophoblastic cells. This barrier could be useful in reducing the maternal oversupply to the fetus. In contrast to other previous work [19] we found an increase in p57 expression in syncytiotrophoblast cells. We hypothesized that p57 expression in GDMd placentas is modulated by the oxidative environment: the oxidative stress would act as an antiapoptotic stimulus rather than a proapoptotic one [22], leading to the persistence of “immature”, circumferential syncytiotrophoblasts, which does not involve creating the normal, polarized, syncytial knot usually seen in normal terminal villi. In this way, the syncytiotrophoblast cytoplasm helps create a physical barrier to the formation of the VSM.

## 5. Conclusions

The empirical results reported herein should be considered in light of some limitations.

The retrospective nature of the study did not allow for the collection of all the precise potential data about the mother’s health and pregnancy. This problem could be easily encompassed by the prospective enrollment of patients and via the use of adequate surveys to explore dietary and lifestyle habits apt to produce effects on pregnancies. We could not enroll insulin-treated patients as there are many different therapeutic schemes and they were not always clearly retrievable by existing databases.

We also acknowledge that the sample size is limited due to limited resources and that the control group is heterogeneous concerning the placental pathology. The latter is due to the adherence to the guidelines regarding the referral of the placentas to the Pathology Laboratory, so placentas delivered from normal pregnancies are not available. At any rate, the cases and controls were matched by the propensity scores. The study was further limited by the absence of previous research studies on this specific topic. Prior research relevant to our thesis is scarce and exquisitely experimental, based on cell cultures isolated from placental tissue; for these reasons, we had to develop a completely new approach, based on placental tissue and immunohistochemistry.

Our work seems to corroborate, not regarding the cultured cells but the placental tissue, the hypothesis that the absence/low expression of hENT1 in endothelial cells in all GDMd placentas may indicate a potential role in microvascular adaptative mechanisms. As the placental microenvironment is extremely complex, many different pathways and metabolic mechanisms likely rely on the alterations found both at cellular and phenotypic levels in GDM. We described a “proliferative”/antiapoptotic pattern (high Ki-67, PHH3, and p57 counts) in the GDMd placentas, which seems to sustain the statistically significant lower number of VSM/mm^2^ found in the GDMd cases when compared to the controls. The combination of the peculiar GDM parameters (absence of hENT1, lower VSM count, high MIB1, PHH3, and p57) could discriminate between GDM (irrespective of morphological features) and non-GDM placentas. Future studies should investigate the expression of hENT1 in non-clinically evident GDM and insulin-treated GDM patients. 

## Figures and Tables

**Figure 1 diagnostics-13-02034-f001:**
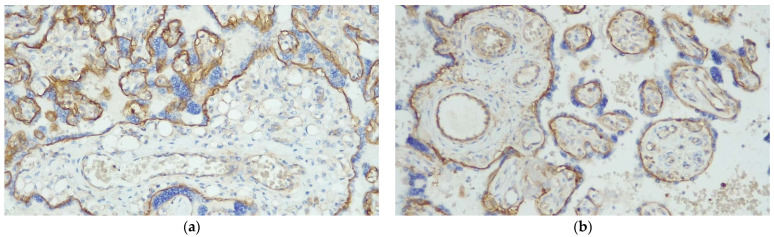
Placental tissue, hENT1 expression. The basement membrane of the villous trophoblast served as a positive internal control. (**a**) GDMd case, hENT 1 expression, score 0 in hPMEC (original magnification 20×); large, “edematous” terminal villi, with centrally located vessels and a continuous layer of syncytiotrophoblasts, which decorates the villi circumferentially. The centrally located vessels are negative for hENT1. VSMs are minimally formed. (**b**) non-GDM case, MVM, hENT 1 expression, score 250 in hPMEC (original magnification 20×). In MVM, the terminal villi are often smaller than usual, often with a “pencil-like” shape (so-called accelerated maturation). Syncytial knots are prominent. The syncytiotrophoblast layer is mainly polarized. All the vessels are positive for hENT1.

**Figure 2 diagnostics-13-02034-f002:**
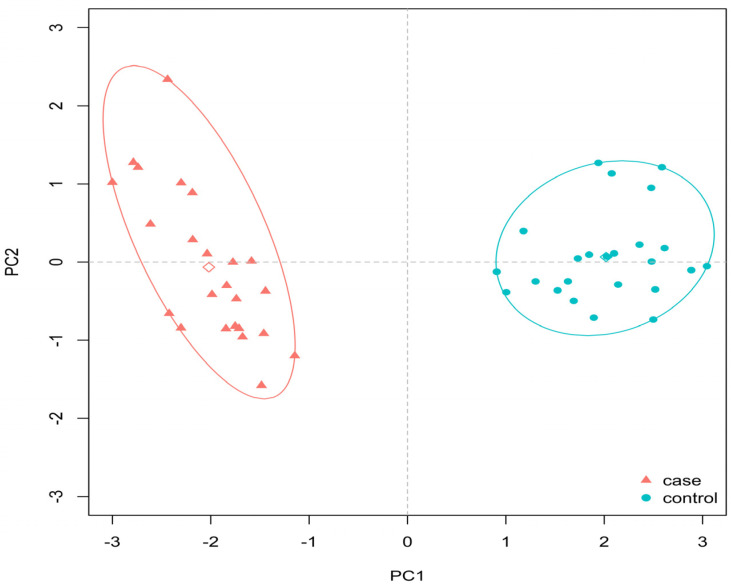
Plot of the first two Principal Components after LASSO feature selection, with elliptic convex hull for cases (GDMd) and controls (non-GDM) patients. The first two components account for 84.21% of the total variability. Empty diamonds represent the mean values (First Principal Component loadings: hENT1 HUVEC = 0.429, hENT1 hPMEC = 0.421, VSM = 0.373, Ki-67 = −0.444, PHH3 = −0.345, p57 = −0.428).

**Figure 3 diagnostics-13-02034-f003:**
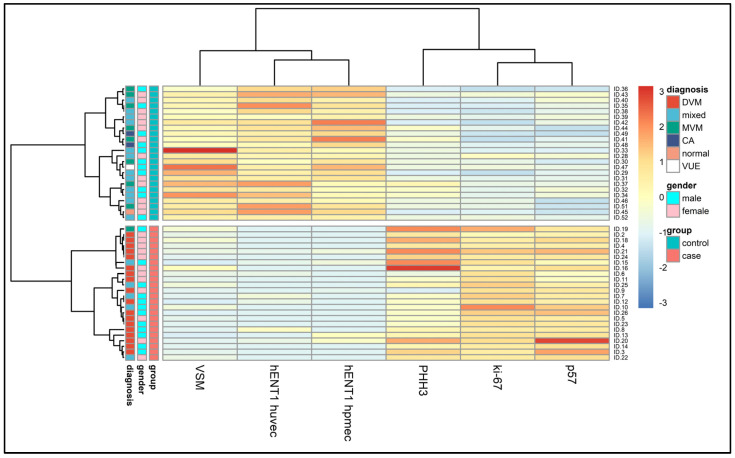
Heatmap of 24 GDMd patients matched with 24 non-GDM patients. After a LASSO-based selection, only six features were retained as informative. Hierarchical clustering was based on Euclidean distance for patients and pairwise correlational distance (1 − |ρ|) for features.

**Table 1 diagnostics-13-02034-t001:** Clinical characteristics of pregnant women, newborns, and deliveries.

Variable	GDMd (Mean ± SD)	Non-GDM (Mean ± SD)	
Mothers			*p*
Age (yrs)	34.62 ± 4.28	33.67 ± 4.69	*0.48*
BMI (kg/m^2^)			
Pre-pregnancy	24.19 ± 4.85	23.98 ± 4.23	*0.87*
At term	27.77 ± 4.32	28.145 ± 4.55	*0.78*
Systolic blood pressure (mmHg)	118.75 ± 11.07	123.75 ± 18.14	*0.25*
Diastolic blood pressure (mmHg)	73.08 ± 9.32	74.67 ± 10.98	*0.60*
OGTT (mg/dL)		Not performed	
Basal glycemia	86.04 ± 9.34	-	
1-h glycemia after glucose load	162.08 ± 29.03	-	
2-h glycemia after glucose load	153.87 ± 25.98	-	
**Newborns**			
Sex (female/male)	12/12	13/11	
Gestational age (weeks)	39.05 ± 1.11	37.95 ± 2.49	*0.056*
Birth weight (grams)	3248.54 ± 481.66	2626.12 ± 568.67	*0.0002*
Placenta/fetus ratio	0.14 ± 0.02	0.14 ± 0.02	*0.87*
APGAR score (5 min)	9.83 ± 0.38	9.21 ± 0.97	*0.007*
**Deliveries**			
Type of delivery			
Vaginal, spontaneous, n (%)	10/24 (41.67%)	11/24 (45.83%)	
Vaginal, induced, n (%)	13/24 (54.17%)	1/24 (4.16%)	
Caesarian, n (%)	1/24 (4.16%)	10/24 (41.67%)	
Operative, n (%)	0/24 (0%)	2/24 (8.34%)	

**Table 2 diagnostics-13-02034-t002:** Morphological characteristics of placentas and immunohistochemical findings.

Variable	GDMd (Mean ± SD)	Non-GDM (Mean ± SD)	*p*
Morphological findings			
Weight (grams)	477.54 ± 75.96	394.91 ± 120.96	*0.007*
Diameter (cm)			
Maximum	17.29 ± 1.89	16.26 ± 1.86	*0.07*
Minimum	15.21 ± 1.61	14.10 ± 2.02	*0.04*
Thickness (cm)			
Maximum	3.48 ± 0.59	3.41 ± 0.82	*0.76*
Minimum	1.73 ± 0.80	1.49 ± 0.78	*0.30*
Cord length (cm)	27.08 ± 7.38	30.85 ± 11.97	*0.10*
Umbilical Cord Index (UCI, number of umbilical veins twists/cm)	0.32 ± 0.12	0.32 ± 0.22	*0.94*
**Histological diagnosis**	**GDMd**	**Non-GDM**	
Number of cases, %	DVM 20 (83.3%)	MVM 12 (50%)	
	DVM + MVM 4 (16.7%)	MVM + DVM 8 (33%)	
		CA 2 (8.3%)	
		VUE 1 (4.2%)	
		Normal 1 (4.2%)	
**Immunohistochemical and morphological study**	**GDMd (mean ± SD)**	**Non-GDM (mean ± SD)**	** *p* **
hENT1 HUVEC	11.25 ± 21.93	194.17 ± 66.52	*<0.0001*
hENT1 hPMEC	10.00 ± 23.03	159.17 ± 54.76	*<0.0001*
Ki-67	19.96 ± 2.83	7.50 ± 2.37	*<0.0001*
PHH3	3.79 ± 2.26	0.96 ± 0.80	*<0.0001*
p57	20.87 ± 4.72	8.45 ± 2.84	*<0.0001*
VSM	3.67 ± 2.96	20.04 ± 8.77	*<0.0001*

## Data Availability

The data are available upon request due to restrictions (privacy). The data presented in this study are available upon request from the corresponding author.

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
