# Peer review of "Gestational Diabetes—Placental Expression of Human Equilibrative Nucleoside Transporter 1 (hENT1): Is Delayed Villous Maturation an Adaptive Pattern?"

_diagnostics, 2023, doi:10.3390/diagnostics13122034_

Round 1

Reviewer 1 Report

Thank you for requesting  to provide a review of this article regarding the placental expression of human equilibrative nucleoside transporter 1 (hENT1) and the possibility of adaptative pattern in the delayed villous maturation. 

   The main purpose of the analysis was to investigate the immunohistochemical expression of hENT1 in human umbilical vein endothelial cells (HUVEC) and hPMEC (human placenta microvascular endothelium cells) in delivered placentas of GDM (gestational diabetes mellitus) patients in dietery treatment.  The main question adressed in the research was if it is possible to evaluate the adaptative pattern in the delayed villous maturation in GDM placentas. The study included the quantitative evaluation of VSM, ki-67, PHH3 and p57 and their correlation with hENT1 and with fetal, placental and maternal characteristics.

   The study is a retrospective observational study, in which 81 placentas were submitted to the Pathology Laboratory for a period of time between April and July 2018. The topic is original and relevant in the field and brings usefull knowledge regarding the subject. A comprehensive search strategy was used. The review methodology was comprehensive with screening and data extraction. When it comes to the methodology used, no specific improvements should be considered from my point of view.

   The conclusions are consistent with the evidence and the arguments presented, and they adress properly to the main question which conducted the analysis.

   The references are appropriate and well suited for this kind of study. 

    The tables used in the article are very precise and explicit and the information written there is very easy to be followed, so no other improvements are required from my point of view.

  Regarding the structure and accuracy of the phrases, the manuscript has well structured information, with supported evidence and well structured phrases.

   The manuscript is original and well defined. The results provide an advance in current knowledge. The results are being interpreted appropriately and are significant, as well as the conclusions.

  The study is correctly designed and the analysis is being performed at high standards, so the data are robust enough to draw the conclusion. Surely the paper will attract a wide readership. 

   To conclude, the article is written in a proper way and brings useful information regarding the subject. However I have some issues to add in the lines below: 

Line 20: in the human, not „in human”

Line 21: placental, not „placenta”

Line 22: the physiological, not „physiological”

Line 27: „:” instead of „.” after „trophoblast”

Line 53: come from, not „from”

Line 55: has been proposed, not „has been proposed by”

Line 275: adaptation, not „adaption”

Line 277: which drives, not „which drive”

Line 353-354: did not allow the collection of all the precise potential data about..., not „did not allow for the collection of precisely all the potential data about...” 

Line 20: in the human, not „in human”

Line 21: placental, not „placenta”

Line 22: the physiological, not „physiological”

Line 27: „:” instead of „.” after „trophoblast”

Line 53: come from, not „from”

Line 55: has been proposed, not „has been proposed by”

Line 275: adaptation, not „adaption”

Line 277: which drives, not „which drive”

Line 353-354: did not allow the collection of all the precise potential data about..., not „did not allow for the collection of precisely all the potential data about...” 

Author Response

Dear Reviewer, Thank you for your comments and the thorough reading of our paper. 

We highly appreciate your accurate and precise work.

We addressed the English language issues you mentioned.

Line 20: in the human, not „in human”. Corrected

Line 21: placental, not „placenta” Corrected

Line 22: the physiological, not „physiological” Corrected

Line 27: „:” instead of „.” after „trophoblast”. I'm sorry, I did not understand if I have to use "instead of"at the beginning of the subsequent sentence, before "hent1", as in line 27 after "trophoblast" there is a full stop. I apologize if I misinterpreted the suggestion.

Line 53: come from, not „from”. "Results" was used as a verb, so I rephrased it as "results in", instead that "results from".

Line 55: has been proposed, not „has been proposed by” Corrected, "by" was canceled.

Line 275: adaptation, not „adaption” Corrected

Line 277: which drives, not „which drive” Corrected

Line 353-354: did not allow the collection of all the precise potential data about..., not „did not allow for the collection of precisely all the potential data about...” rephrased as suggested.

We hope to have satisfied the Reviewers’ and Editorial concerns with the revised version of the manuscript.

We would like to thank you for your attention and we are looking forward to hearing from you.

Best Regards,

On behalf of all Authors

Cinzia Giacometti

Reviewer 2 Report

The manuscript submitted by Giacometti et al. is of interest and is within the scope of this Journal. The manuscript is well written and has solid results. The conclusions are well established and supported by the results. In this sense, the discussion provides an adequate and scientifically coherent approach.

Only this reviewer has two minor issues that the authors can easily resolve. First of all, figure 1 should be better described from the histological point of view. Authors must describe each of the parts of the placental villus precisely. Figure 1 should have a better figure legends. Secondly, the authors must statistically justify the sample size, the authors must calculate the statistical potential.

A simple question and suggestion is that paragraph 6 should be discussed in the discussion section.

Authors should improve the use of English grammar.

Minor editing of English language required

Author Response

Dear Reviewer, Thank you for your comments and the thorough reading of our paper. 

We highly appreciate your accurate and precise work.

We addressed the issues you mentioned in your Review.

We hope to have satisfied the Reviewers’ and Editorial concerns with the revised version of the manuscript.

First of all, figure 1 should be better described from the histological point of view. Authors must describe each of the parts of the placental villus precisely. Figure 1 should have a better figure legends. The caption of Figure 1 has been improved, following your suggestions.

Figure 1. Placental tissue, hENT1 expression. The basement membrane of the villous trophoblast served as a positive internal control. a) GDMd case, hENT 1 expression, score 0 in hPMEC (original magnification 20x); large, “edematous” terminal villi, with centrally located vessels and a continuous layer of syncytiotrophoblast, which decorates the villi circumferentially. The centrally located vessels are negative for hENT1. VSMs are minimally formed. 1B) non-GDM case, MVM, hENT 1 expression, score 250 in hPMEC (original magnification 20x). In MVM, the terminal villi are often smaller than usual, often with a “pencil-like” shape (so-called accelerated maturation). Syncytial knots are prominent. The syncytiotrophoblast layer is mainly polarized. All the vessels are positive for hENT1.

Secondly, the authors must statistically justify the sample size, the authors must calculate the statistical potential. We checked the test power (paragraph 2.1.1. Pre-processing, line 183). The results are reported in the Results section, lines 233-235 "The resulting sample size allowed a test power of  0.95 with respect to a Cohen’s effect size  0.8 and a significance level 0.05."

A simple question and suggestion is that paragraph 6 should be discussed in the discussion section.

We moved paragraph 6 in the discussion section.

Authors should improve the use of English grammar. We double-checked English grammar. 

We want to thank you for your attention, and we are looking forward to hearing from you.

Best Regards,

On behalf of all Authors

Cinzia Giacometti